# Cognitive Aging in Older Breast Cancer Survivors

**DOI:** 10.3390/cancers15123208

**Published:** 2023-06-16

**Authors:** James C. Root, Yuelin Li, Elizabeth Schofield, Irene Orlow, Elizabeth Ryan, Tiffany Traina, Sunita K. Patel, Tim A. Ahles

**Affiliations:** 1Neurocognitive Research Lab, Department of Psychiatry and Behavioral Sciences, Memorial Sloan Kettering Cancer Center, New York, NY 10065, USA; liy12@mskcc.org (Y.L.); schofiee@mskcc.org (E.S.); ryane1@mskcc.org (E.R.); ahlest@mskcc.org (T.A.A.); 2Molecular Epidemiology Laboratory, Department of Epidemiology and Biostatistics, Memorial Sloan Kettering Cancer Center, New York, NY 10065, USA; orlowi@mskcc.org; 3Department of Medicine, Memorial Sloan Kettering Cancer Center, New York, NY 10065, USA; trainat@mskcc.org; 4Departments of Population Science and Supportive Care Medicine, City of Hope Comprehensive Cancer Center, Duarte, CA 91010, USA; spatel@coh.org

**Keywords:** cancer, cognition, aging, deficit accumulation

## Abstract

**Simple Summary:**

The impact of cancer and cancer treatment on longer-term cognitive aging trajectories is currently unknown. Cancer and cancer treatment accelerate biological aging and may accelerate cognitive decline in older individuals. We compared younger and older breast cancer survivors with younger and older non-cancer controls using standard cognitive measures to estimate age- and cancer-related differences over time. We found the expected inverse association of age with cognition in both groups, lower learning and memory performance for survivors as a whole, and more prominent differences in learning and memory as well as attention, processing speed, and executive function in younger survivors, i.e., those under 75 years of age. These differences were similar to trends across the age span in deficit accumulation, with larger differences in younger survivors that may indicate a mechanism of cognitive aging more generally, and in younger survivors specifically.

**Abstract:**

Background: Cancer and cancer treatments may affect aging processes, altering the trajectory of cognitive aging, but the extant studies are limited in their intervals of assessment (two–five years). We studied the cognitive performance of a cohort of survivors and controls aged from 60 to 89 years utilizing cross-sectional cognitive performance data as an indicator of potential aging trajectories and contrasted these trends with longitudinal data collected over two years. Methods: Female breast cancer survivors who had been diagnosed and treated at age 60 or older and were 5- to 15-year survivors (N = 328) and non-cancer controls (N = 158) were assessed at enrollment and at 8, 16, and 24 months with standard neuropsychological tests and comprehensive geriatric assessment. Results: A cross-sectional baseline analysis found the expected inverse association of age with cognition in both groups, with survivors performing lower overall than controls in learning and memory (LM). Younger survivors, i.e., those under 75 years of age, exhibited lower performance in both LM and attention, and processing speed and executive function (APE), compared to controls, with no differences being observed between older survivors and controls, which tracked with deficit accumulation trends. Conclusion: Cognitive differences between the survivors and controls for the LM and APE domains were prominent in younger survivors, as was deficit accumulation, suggesting a mediating effect on cognition. Deficit accumulation may represent a modifiable risk factor in cancer survivorship that may be targeted for prevention and intervention.

## 1. Introduction

The literature examining cancer-related cognitive dysfunction (CRCD) in cross-sectional designs [1,2,3,4,5], as well as in prospective, longitudinal studies that report effects in short-interval follow-up assessments following treatment, is now a fairly extensive [6,7,8,9,10,11,12]. Longitudinal studies also aim to examine whether cancer and cancer treatment alter biological and cognitive aging trajectories, potentially exacerbating the decline associated with normal cognitive aging. While previous research is suggestive of this, there are limitations to how these potential trajectories have been examined. The existing studies typically follow survivors with repeated testing over a relatively short window around the time of treatment and after two–five years, in which (1) little cognitive aging is expected, (2) practice effects due to repeated testing lead to the appearance of increasing cognitive performance with age, and (3) selective attrition, in which the most robust of the sample continue on in the study, obscures expected cognitive aging trends.

Long-term, there is evidence of persistent CRCD in cross-sectional studies at 5 [13], 10 [4], and 20 [14] years post-treatment, although little is known regarding why or how cognition changes, i.e., the dynamics of the rate and magnitude of change over developmentally meaningful intervals. This is an important gap since etiologies of cognitive dysfunction around the time of treatment may be distinct from potentially modifiable factors at longer intervals post-treatment. Aging trajectories may be influenced by multiple factors, with a growing body of research examining deficit accumulation and its effects. Deficit accumulation is characterized by comorbidity burdens, polypharmacy, social detriments of disease (e.g., smoking, obesity), psychological disturbance, and functional limitations/declines in activities of daily living [15]. A relationship between frailty and cognitive impairment has been reported in the geriatric literature [16,17,18]. Cancer treatment is associated with an acceleration in the accumulation of other comorbidities and deficits over longer intervals [19,20,21,22], and recent work from our group has identified an association of deficit accumulation with cognitive function in breast cancer patients [23,24,25]. In an earlier study, we found that the comorbidity burden was higher in cancer patients and associated with baseline cognitive function prior to adjuvant treatment [23]. In parallel, recent research found that longer-term survivors had higher deficit accumulation and worse cognition compared with controls [24], and that cognitive differences were largely mediated by increased deficit accumulation in survivors [25]. This work focused on cognitive and deficit accumulation trajectories over a two-year interval and, therefore, was limited for the same reasons discussed above, i.e., repeated testing and short period of follow-up, and leaves open the dynamics of longer-term trajectories and association with deficit accumulation. Since deficit accumulation may be modifiable with appropriate preventative measures or managed with concurrent treatment, a better understanding of deficit accumulation and its long-term impact on cognition may improve outcomes for long-term survivors.

There are clear impediments to examining these longer-term outcomes longitudinally. Following survivors over a 10–20-year interval is logistically unfeasible and, similar to the limitations noted above for shorter interval studies, practice effects and selective attrition will distort the expected course of cognitive decline associated with age and survivorship. Practice effects are inherent in repeated cognitive testing [26] and have a tendency to obscure age-related decline as a result of improving performance, given that the subject will have had previous experience with the measures [27,28]. The presence of practice effects may be especially problematic for the study of cognition in survivorship given our previous findings of initial learning and attention deficits in breast cancer survivors and evidence that repetition within and across testing sessions leads to steeper improvements in performance in survivors than in controls [29]. As an alternative approach, the methods and analysis presented herein stem from aging researchers, whose primary focus is on normal cognitive aging across the lifespan, and harness cross-sectional cognitive data to capture developmental changes over longer intervals. Previous work in normal cognitive aging has found that age-associated longitudinal and cross-sectional cognitive trajectories disagree, with the former exhibiting increases in performance with age and the latter exhibiting decreases [27]. While the cross-sectional trend is intuitively more explicable, concerns have been raised about potential cohort effects in different age groups that may be confounded with age. These include early developmental variables, e.g., educational opportunities, nutritional access, acculturation, etc., that can result in differences in cognition between generational cohorts that confound the effects of age [30]. While a significant and robust finding in earlier work, there is increasing evidence that cohort effects, this pattern of increasing cognitive scores in successive generational cohorts over the 20th century, has diminished as environmental factors have stabilized in developed countries [31,32], including the United States [33]. Research from the literature on aging that disambiguates cohort and aging effects suggests that cohort effects have decreased [27,34] and, with the inclusion of a non-cancer control group at matched ages, any cohort effects in survivors will be controlled for by contrasts with control participants who are from the same age cohort. Given the confounding effects of repeated longitudinal assessment on cognitive trajectories, selective attrition, and the logistical challenges in following individuals over aging intervals of interest (5–20 years), examining cross-sectional cognitive data across the lifespans of the subjects and evaluating differences between survivors and controls may serve as a useful and accurate proxy indicator of how a history of cancer may alter cognitive aging trajectories.

The data reported here were collected as part of a collaboration between Memorial Sloan Kettering Cancer Center and City of Hope (PIs: Ahles; Hurria) on a cohort study that assessed cognition in controls and breast cancer survivors (age 60 or greater) who were all at least five years post-treatment and were followed prospectively over two years with four timepoints at eight-month intervals [35]. In this secondary analysis, we examined data derived from the first time-point across a longer 5–15-year interval, in four quartiles, as a proxy indicator of how cognitive trajectories might be altered given a history of cancer and cancer treatment. We then compared these trends to longitudinally collected data over shorter intervals (two years) to examine the potential distorting effects of repeated cognitive testing. Finally, we examined the association of deficit accumulation with cognitive differences between survivors and non-cancer controls across the age span.

## 2. Materials and Methods

**Participants.** Breast cancer survivors were identified through the survivorship clinics at Memorial Sloan Kettering Cancer Center (MSK) and City of Hope Comprehensive Cancer Center (COH), and participants were supplemented at each site by recruitment through the Army of Women. Survivors were eligible if they were diagnosed with stage 0-III breast cancer, treated at age 60 or above, were from 5- to 15-year disease-free survivors at the time of enrollment, and provided informed consent. Survivors were excluded based on the following criteria: score of 11 or greater (indicating risk of dementia) on the Blessed Orientation-Memory-Concentration (BOMC) Test, previous history of cancer (except non-melanoma skin cancer), treatment with chemotherapy for non-cancer conditions, neurobehavioral risk factors, including history of a neurologic disorder (e.g., seizure or dementia), alcohol/substance abuse, head trauma requiring hospitalization or evidence of structural brain changes on imaging, or severe psychiatric disorder (e.g., schizophrenia, bipolar disorder). Female non-cancer controls who met the same inclusion criteria (except for diagnosis of cancer) and exclusion criteria were recruited through community advertisement and the Army of Women. Survivor and control groups were frequency matched so that mean age and education were equivalent, but not on a one-to-one basis. All methods and procedures were approved by the institutional review boards of MSK and COH. Toward the end of the study, the age at diagnosis was lowered to 55 to increase the number of survivors who had been treated with chemotherapy. Twenty-three participants who had been treated between 55 and 60 were recruited and, thus, contribute to the overall cohort of N = 486 women.

**Measures**. Assessments occurred at enrollment and at 8-, 16-, and 24-month follow-ups, and, thus, the study follows a longitudinal observational design with two cohorts, cancer survivors and matched non-cancer controls. The assessment battery included standardized neuropsychological tests, self-report of cognitive function, and components of the Comprehensive Geriatric Assessment [36] which were used to calculate the Deficit Accumulation Frailty Index (DAFI). The neuropsychological measures were categorized into domains based on previous studies [36] and the clinical judgment of the neuropsychologists involved with the study (JCR, ER, SP), informed by a factor analysis.

Each test score was first standardized (z-score) according to the healthy control group, and then the mean of the standardized scores within the domain was calculated for each participant. Individual test scores were checked for deviation from a normal distribution. For those that differed, the Box–Cox algorithm [37] was used to determine a suitable power transformation prior to domain score calculations. Below are the administered tests categorized by domain:

**Attention, Processing Speed, Executive Function**: Digit Symbol [38]; Trail Making A and B [39]; DKEFS Color-Word Naming [40]; NAB Digits Forward and Backward [41]; NAB Driving Scenes [41].

**Learning and Memory:** NAB List Learning [41]: Trial 1, Semantic Clustering, List A Immediate, List A Delayed, Long Delay, List B Immediate, and New Recognition Index; Logical Memory Part 1 and 2 [42].

**DAFI Score**: Measures used to calculate the DAFI score assessed: (1) Functional Status: Instrumental Activities of Daily Living (IADL) Subscale of the Multidimensional Functional Assessment Questionnaire [43]; Medical Outcomes Study (MOS) Physical Health, Social Limitations, and Social Support Scales [44]; Karnofsky Self-Reported Performance Status Scale [45]; Self-report of the number of falls in the last 6 months; (2) Comorbidity: Physical Health Section Older American Resources & Services Questionnaire (OARS) [43] and a single sum of the 14 items; (3) Depression: Center for Epidemiological Study—Depression [46]; (4) Anxiety: Spielberger State Anxiety Inventory [47]; (5) Fatigue: Fatigue Symptom Inventory [48]. Finally, the timed Up and Go test was administered [49].

The DAFI score was quantified as a score ranging between zero and one based on up to 44 possible frailty indicators, as described by Cohen et al. [50]. For each indicator (e.g., limited ability to climb one flight of stairs, a diagnosis of arthritis) the participant scored a zero, one, or two based on whether the indicator showed absent, intermediate, or most adverse risk, respectively. The deficit accumulation frailty index (DAFI) score was then calculated as the sum of these individual indicator scores divided by the maximum possible score. In cases where an indicator variable was missing, the item was excluded from both numerator and denominator. The score was calculated for all participants for whom at least 35 indicators were assessed (2 participants were excluded). Continuous DAFI scores were than used to classify participants as robust (DAFI < 0.2), pre-frail (0.2 DAFI ≥ 0.2 < 0.35), or frail (DAFI ≥ 0.35). Since we were interested in the relationship between deficit accumulation and cognition, self-report of cognitive function was not included as a frailty indicator. Additionally, to utilize the same criteria for survivors and controls, breast cancer history was not included as an indicator (3% of the sample had a history of another type of cancer (e.g., skin cancer), which was included as a frailty indicator).

**Statistical Approach:** We fitted what is known in the literature as a ‘varying-intercepts, varying-slopes’ model [51] in a Bayesian framework. The full model equations are summarized in the Appendix A, where we use plots to visualize the varying intercepts and slopes, and what they represent in the context of this study (see Appendix A). Next, in the Supplement, we cover the model equations to introduce readers to the notation commonly used in Bayesian statistics. We then describe selected model parameters in relation to this paper. Raudenbush and Bryk [52] were among the first to introduce social scientists to the idea of modeling varying slopes and intercepts in hierarchical data. Further readings on varying intercept and slope models are provided, as well as introductory Bayesian textbooks for non-technical readers. Briefly, the model is divided into two levels. In level 1 of the model, longitudinal assessments for the *i*th person are first summarized as an intercept $\alpha_{i}$ and a slope $\beta_{i}$. This can be conceptualized as distilling each study participant’s four longitudinal assessments into one intercept and one slope, from each of the 328 survivors and 158 controls. In level 2 of the model, these varying intercepts and varying slopes are further analyzed. The intercepts are expressed as a quadratic function of chronological age at enrollment, and as a continuous variable, and are centered at age 72.5 (average age of the entire sample). The control and survivor cohorts follow two separate quadratic aging trends. The varying slopes are modeled as a function of 4 age quartiles at baseline. We used the categorical age cohorts at enrollment to fit the varying slopes, rather than the continuous chronological age, because of specific research questions (i.e., whether practice over time differed between survivors and controls in these baseline age cohorts), and because we did not want to count time twice (varying slopes were already fitted with time). Additional technical details on the model can be found on an online data repository (https://github.com/bayesnp/RandomSlopesIntcpts, accessed on 22 May 2023), including a worked example with syntax codes.

It should be noted that the person-specific intercepts represent each person’s score at enrollment—the cognitive performance at the first assessment occasion. This first assessment is what Salthouse [53] characterizes as the *cross-sectional* score, while the repeated assessments represent the *longitudinal* score. This terminology is in part attributed to a well-documented finding in cognitive aging research, which is that cognitive performance often improves over repeated test administrations. This repeated test exposure effect contains the practice effect, which confounds within-person cognitive changes. By contrast, the first assessment is less prone to the practice effect, and is presumably absent in participants who have never encountered the test before. In the literature, the first assessments are typically used for *between-subject* comparisons, such as cognitive performance between chronological age groups. The longitudinal scores are analyzed separately to minimize confounding. In this paper, we follow this convention and refer to the varying intercepts as the cross-sectional scores and the varying slopes as the longitudinal scores, despite the unfortunate confusion this may cause, especially to readers who use them for study design rather than outcomes. We improve on the conventional approach by offering two noteworthy features in the proposed model. First, the cross-sectional aging trend is modeled continuously over chronological age, which improves on the categorical age bins frequently used by aging researchers, and the practice effects are modeled simultaneously, which improves on the separate analyses in Salthouse [53]. Second, the covariance Ω allows slopes and intercepts to be correlated. For example, a positive correlation indicates that individuals who have a higher cross-sectional cognitive performance at enrollment also tend to show a greater practice effect over time. This correlation is not available when the cross-sectional and longitudinal data are analyzed separately.

Bayesian computation was done using the rstanarm package (version 2.21.3: https://mc-stan.org/docs/reference-manual/index.html, accessed on 22 May 2023) in R version 4.2.2. Convergence of the simulations was evaluated by the $\hat{R}$ ≤ 1.01 diagnostic metric [54], achieved in all models with 4 chains of 60,000 iterations each, 10,000 of which were omitted as warm up iterations and using a thinning interval of 5. Parameter estimates and their 95% Highest Density Intervals (HDI) were sought. Source code files for model fitting and for plotting key figures are available in the online repository.

## 3. Results

Table 1 summarizes the demographic and treatment characteristics of the sample. The analytic sample included 328 cancer survivors and 158 cancer controls, as described previously [25]. The sample was largely white (85%), non-Hispanic (88%), and had undergraduate education or greater (59%). The ages in this study ranged from 60 to 89 years, with an average of 72.5 years. Recruitment for this study was targeted so that approximately 50% had a history of treatment with chemotherapy or no chemotherapy. The majority of the patients were ER positive (80%), PR positive (65%), and HER2 negative (88%), and had been treated with endocrine therapy (75%). Twenty-five percent of the survivors were actively receiving endocrine therapy when enrolled in the study.

Table 2 summarizes the parameter estimates for the attention, processing speed, and executive function (APE) and Learning and Memory (LM) models. As explained in the Appendix A, the intercepts are fitted as a quadratic function of chronological age. For controls, their quadratic age function has the shape $\gamma_{00}+\gamma_{01}{Age}_{i}+\gamma_{02}{Age}_{i}^{2}$, where $\gamma_{00}$ represents the overall intercept, $\gamma_{01}$ the linear age term, and $\gamma_{02}$ the quadratic age term. The survivors have a separate quadratic age function, with $\gamma_{03}$, $\gamma_{04}$, and $\gamma_{05}$, representing, respectively, the differences between survivors and controls in the overall intercept, the linear age term, and the quadratic age term.

For the APE, there are three statistically discernible terms, for which the HDI excludes the null. The first is a reliable linear term, chronological age at enrollment ($\gamma_{01}$: −0.047, Bayesian 95% HDI: −0.065, −0.029). The age effect translates to a worsening APE in the control cohort by 0.47 z-scores per decade of aging, what Cohen would consider a ‘medium’ effect [55] in psychological research. Note that this term is derived from the *cross-sectional* scores made at the first assessment occasion and, thus, is not confounded by the practice effect. The second statistically reliable term is an average practice effect over months ($\gamma_{10}$: 0.004, HDI: 0.0004, 0.007), showing an estimated *longitudinal* increase of 0.004 in APE per month (0.96 increase over the course of 24 months). This estimated within-person change is confounded by the practice effect. The third statistically discernible term shows that the oldest age quartile (77–89-year-olds) had a reliably lower longitudinal improvement as compared to the youngest age quartile (60–68-year-olds). The survivors showed subtle differences from controls, shown in the coefficients veering in the anticipated directions, but the HDIs did not exclude the null. For instance, survivors had a lower age intercept ($\gamma_{03}$: −0.130, HDI: −0.279, 0.020, did not exclude the null). Survivors had a slightly lower aging quadratic term by 0.1 z-scores per decade of aging ($\gamma_{05}$ = -0.01, HDI: −0.004, 0.002), and the lower bound suggested a potentially greater gap, although the HDI did not exclude the null. The random effect showed an estimated correlation of 0.16 between the random intercepts and slopes, indicating that a higher APE score at enrollment was associated with a greater longitudinal slope over 24 months.

A similar pattern was observed in the learning and memory domain scores, where reliable fixed effects were found in the age linear term (−0.048, HDI: −0.069, −0.028) and longitudinal changes over months (0.019, HDI: 0.013, 0.025). The age effect translates to a worsening Learning and Memory score by 0.48 z-scores in the control cohort per decade of aging, comparable to the 0.47 in the APE. Survivors had a lower age intercept that excluded the null (−0.205, HDI: −0.376, −0.034). There was a 0.27 correlation between the random intercepts and slopes, slightly greater than the 0.16 correlation in the APE domain.

Figure 1a provides a visual explanation for the APE model, where the model-estimated APE scores are plotted over chronological age. In the left panel, the solid lines represent the model-estimated, cross-sectional APE scores for the survivors and controls. Note that these are derived from the first assessments and, thus, are not affected by the practice effect. The most noteworthy visual feature of Figure 1a appears to be a gap between the solid curves between survivors and controls in individuals younger than (approximately) 72 years of age, beyond which the credible intervals begin to converge. To further examine this gap, we plotted, on the right panel, the average difference between the survivors and controls (in the solid line) and its 95% credible intervals (shaded areas). At age 65, the survivors are estimated to have a reliably lower APE performance than controls by 0.31 z-scores (95% HDI: −0.52, −0.08), which excludes the null. The arrow shows that the shaded credible intervals begin to cross the null at age 71.8, where group contrasts are no longer significant. This subtle difference is obscured in Table 2 because the age^2^ × survivor interaction term shows no overall difference in the age quadratic trends between the survivors and controls ($\gamma_{05}$ = −0.01, HDI: −0.004, 0.002). However, Figure 1a shows that the lower quadratic coefficient by −0.01, and the HDI skewing towards a lower bound of −0.004, manifests in a subtle but discernible gap of worse APE scores in survivors younger than 72. Figure 1a also indicates that the oldest age quartile (77–89-year-olds) has a reliably lower longitudinal improvement as compared to the youngest age quartile (60–68-year-olds), which corroborates the statistically reliable $\gamma_{12.4}$ term in Table 1.

Figure 1b shows a similar overall pattern in Learning and Memory, where cross-sectional cognitive performance declines with chronological age. The longitudinal slopes in the filled circles have a visibly pronounced upward increase for all of the age cohorts, appearing to be steeper than those in APE, and even the oldest age quartile retains a steep slope. The right panel shows that, between 64.5 and 74.2 years of age, a discernible gap at 95% posterior confidence is found between the survivors and controls, indicating a significantly lower survivor performance than for controls over this age range; similar to the APE analysis, the performances between groups begin to overlap as individuals age, indicating no significant difference between survivors and controls. The credible interval is somewhat wide for age 64.5 and younger, in part because of relatively sparse data coming from only 16 controls and 27 survivors for this age range.

Figure 1c shows the same model applied to scores on the Deficit-Accumulation and Frailty Index (DAFI). The plot on the left shows that cancer survivors have a greater deficit accumulation than non-cancer controls across the chronological age span. Similar to the pattern seen in Figure 1a,b, Figure 1c demonstrates that the gap in deficit accumulation at enrollment appears to be primarily in the younger age range, and the plot on the right shows that its credible interval crosses the null at age 69.8. Deficit accumulation also appears to be greater in older cancer survivors. However, the average gap plot indicates that the difference is not statistically discernible at 95%, in part due to the relatively small number of older participants. The filled circles show that survivors in the youngest age quartile continue to accumulate deficits over the 24-month duration of the study, while the deficit accumulation appears to subside in the older survivor quartile groups. Interestingly, controls in the oldest age quartile continue to accumulate deficits.

## 4. Discussion

In this analysis, we examined baseline, cross-sectional performance differences between survivors and controls over a 30-year age span as a proxy indicator of how cancer and cancer treatment might affect cognitive aging trajectories. We have taken this approach to sample potential cognitive aging trends over meaningful aging intervals without the effects of practice and selective attrition from repeated testing, as well as to demonstrate the distorting effects of repeated longitudinal testing on cognitive trajectories.

As expected, the longitudinal trajectories suggest an improving cognitive performance at each successive timepoint, driven by practice effects due to repeated test exposure, in contrast to cross-sectional, baseline trends that indicate an inverse association of age and cognition. The longitudinal analysis resulted in two counterintuitive findings regarding age and deficit accumulation and their effect on cognition: increasing age and increasing deficit accumulation, accurately reflecting increasing deficits over time, become artifactually associated with improving cognition. Both associations are at odds with what is known of the influence of age and deficit accumulation on cognition. The literature on aging indicates that advancing age exerts one of the most robustly negative effects on cognition [53]. Likewise, deficit accumulation and frailty also demonstrate an inverse association with cognition in older adults [16,17,18]. Repeated testing obscures both the expected cognitive aging trajectories and associated mediators of cognitive aging, in this case deficit accumulation. The argument could be made that any practice effects would be shared equally between groups and that, in contrasting survivors and controls, these effects are subtracted away, leaving the expected effects of cancer and cancer treatment. We note that practice effects are not equal between domains or age bands and, in previous work, we have found that survivors benefit more from repeated exposure both within and across assessment time-points [29]. In this analysis, for LM, a strong practice effect is notable across all age bands, in contrast to a weaker and declining practice effect for APE with increasing age. Our analysis also found a significant association of baseline performance with the magnitude of the practice effect. This heterogeneity between domains, ages, and baseline performance effects suggests that attempts to model or account for practice effects in any analysis will be methodologically difficult, as this would require the baseline-, age-, and domain-specific modeling of practice.

Similar problems were encountered by researchers studying normal aging and cognition—repeated testing that aimed to establish cognitive trajectories associated with age led to the appearance of improvement in cognition with age. When a quasi-longitudinal (same cohort/different age) design was used, trajectories associated with aging followed the expected cross-sectional trends of declining cognition with age that cannot be explained by a generational cohort effect [27]. The contrast of longitudinal and cross-sectional data from the above work is strikingly similar to the longitudinal and cross-sectional trends that we see in this study—longitudinal improvements versus cross-sectional declines. Examining the first timepoint cross-sectionally across age allows us to see cognitive differences with age in both groups, as well as differences in these trends that are specific to survivors and controls. In both the LM and APE plots, the survivors experience an early decline in cognition at a younger age range and exhibit a declining but flatter slope than controls with increasing age. The survivors’ and control slopes finally meet and continue to overlap from approximately age 70 through to the highest age range. This pattern, while a proxy for true longitudinal trends associated with aging and survivorship, was unexpected given the previously theorized trajectories associated with cancer and cancer treatment. Both a phase shift trajectory, in which deficits persist but parallel cognitive decline in women without a cancer history, and an accelerated aging trajectory, in which there is a steeper slope of cognitive decline with age, have been hypothesized [56]. The phase shift trajectory would be expected if the primary effect on cognition is assumed to be cancer development and treatment, with little recovery in the near- and long-term, creating a new baseline that now parallels similarly aged individuals with no cancer history, both declining equally with age. In contrast, an accelerated aging trajectory might be expected if either (a) the initial effect of cancer treatment leads to a cascade of biologic events, i.e., deficit accumulation, that cause continued cognitive decline with aging, or (b) if a given treatment may not be sufficient to immediately affect cognitive function but may produce a delayed effect as aging continues. Instead, our analysis found evidence of early decline in younger survivors, with controls approaching similar declines in cognition at older ages. While we have demonstrated cognitive differences between survivors and controls in this sample as a whole in previous work [35], the data examined here suggest that it is the younger survivors who contribute most to these observed cognitive differences.

The gaps in deficit accumulation also appear to be greatest for survivors younger than 70. This suggests that gaps in cognition are perhaps associated with gaps in deficit accumulation, and cancer and cancer treatment contribute to the additional deficits in younger cancer survivors as compared to controls. We have previously established a mediating role of deficit accumulation in cognitive dysfunction in survivors in the sample as a whole [25]. To the extent that this pattern is driven by deficit accumulation, this may suggest that younger survivors experience an early increase in deficits associated with cancer and treatment, i.e., comorbidity burden, polypharmacy, social detriments of disease (e.g., smoking, obesity), psychological disturbance, and functional limitations/declines in activities of daily living, compared to individuals without a history of cancer that, in turn, lead to an early decline in cognition. At younger ages, survivors may be most unique from controls for the fact of cancer diagnosis, treatment, and associated deficits. As controls age, comorbidities and functional limitations accrue which are related to normal aging and other etiologies outside of cancer and treatment. With time, and with the normal process of aging and deficit accumulation, absent a history of cancer, older controls accrue deficits at similar levels to those of younger and older survivors, with cognitive performance then converging as a result. This may partly explain why neither a phase shift nor an accelerated aging trend was found. Survivors accrue deficits earlier, but the slope of accumulation is relatively more flat in contrast to controls who steadily accrue increasing deficits with age, converging with survivors.

As with any research, this report is subject to limitations. This is a secondary analysis that was not originally intended in the original proposed study. That proposal intended to examine altered cognitive trajectories using the longitudinal timepoints over a two-year period. Given the difficulties imposed by repeated testing and the relatively short timeframe proposed, this secondary analysis sought to estimate aging trends from only baseline cognitive performance over a longer interval. While we note that generational cohort effects are of less concern given evidence that these effects have diminished, we cannot rule out any cohort effect on our analysis as a whole. As an example, tobacco use has steadily declined in successive generations and previous nicotine exposure has been associated with a potentially protective effect on cognition in APOE4+ carriers [35]. With the addition of control data from equivalent ages/cohorts, we can, however, control for any cohort effects that would influence differences in cognition between survivors and controls. Additionally, work from Salthouse et al. found little effect of cohort effects on cognition, with quasi-longitudinal and cross-sectional trends returning similar trajectories over age.

## 5. Conclusions

This analysis highlights the distorting effects of longitudinal cognitive testing on expected cognitive trajectories as a result of repeated exposure to cognitive measures. We have introduced an analysis informed by normal cognitive aging research using cross-sectional data as a proxy indicator of cognitive aging more generally, and in this case, of differences in cognitive aging associated with a history of cancer and cancer treatment. The additional analysis of deficit accumulation reveals similar trends in deficit accumulation and cognition, and suggests a potential link between the two, specifically in younger survivors. To the extent that deficit accumulation is implicated in cognitive decline in survivorship, this would suggest one potentially modifiable risk factor which could be identified for monitoring, prevention, and intervention.

## Figures and Tables

**Figure 1 cancers-15-03208-f001:**
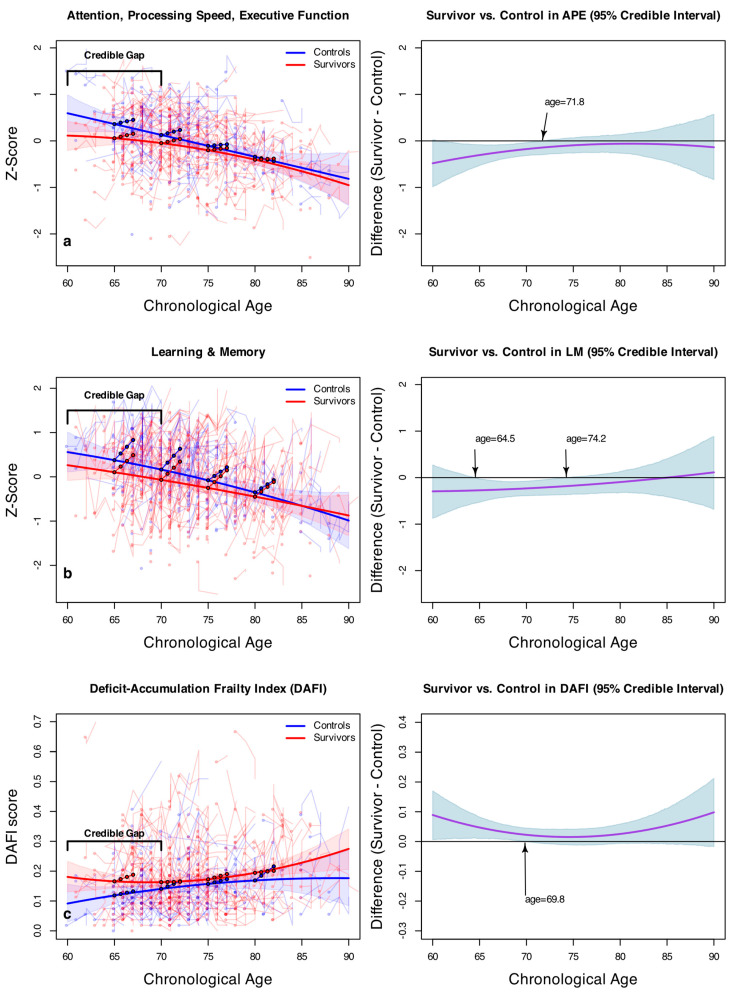
Model-estimated APE (**a**), LM (**b**), and DAFI (**c**) scores as a function of chronological age (**left**) and the expected differences between cancer survivors and controls (**right**), with 95% posterior credible intervals. In the left panels, solid lines represent the model-estimated cross-sectional scores for survivors and controls. Filled circles represent the model-estimated longitudinal changes over 24 months in the age quartiles. Added in opaque colors are the 95% credible intervals for the solid lines and the observed scores for each study participant over the course of up to 24 months.

**Table 1 cancers-15-03208-t001:** Participant characteristics by treatment group.

		No. of Patients (%)			
Characteristic	Overall	Chemo	No Chemo	Control	*p*-Value
	(*n* = 486)	(*n* = 160)	(*n* = 168)	(*n* = 158)	
**Age**, M (SD)	72.5 (5.8)	70.9 (5.1)	74.6 (5.9)	71.7 (5.8)	<0.001
**Race**					
White	411 (85%)	129 (81%)	141 (84%)	141 (89%)	0.320
Black	37 (8%)	16 (10%)	15 (9%)	6 (4%)	
Asian/PI	17 (3%)	8 (5%)	5 (3%)	4 (3%)	
Other	14 (3%)	5 (3%)	5 (3%)	4 (3%)	
Missing	7 (1%)	2 (1%)	2 (1%)	3 (2%)	
**Ethnicity**					
Hispanic	48 (10%)	18 (11%)	12 (7%)	18 (11%)	0.338
Non-Hispanic	429 (88%)	138 (86%)	153 (91%)	138 (87%)	
Missing	9 (2%)	4 (3%)	3 (2%)	2 (1%)	
**Education**					
Less than college	195 (40%)	67 (42%)	71 (42%)	57 (36%)	0.420
College or more	289 (59%)	92 (58%)	96 (57%)	101 (64%)	
Missing	2 (0%)	1 (1%)	1 (1%)	0 (0%)	
**Smoking Hx**					
Yes	221 (45%)	85 (53%)	74 (44%)	62 (39%)	0.036
No	263 (54%)	74 (46%)	93 (55%)	96 (61%)	
Missing	2 (0%)	1 (1%)	1 (1%)	0 (0%)	
**Endocrine Therapy**					
Ever	234 (75%)	110 (72%)	124 (77%)	NA	0.298
At Assessment 1	80 (25%)	52 (34%)	28 (17%)	NA	<0.001
**Cancer Characteristics**					
ER Positive	237 (80%)	111 (73%)	126 (88%)	NA	0.001
PR Positive	190 (65%)	84 (55%)	106 (76%)	NA	<0.001
HER2+ (FISH)	32 (12%)	25 (17%)	7 (6%)	NA	0.006
Tumor size (cm)	1.7 (1.4)	2.2 (1.5)	1.2 (1.2)	NA	<0.001
Years since DX	8.0 (2.7)	8.1 (2.7)	8.0 (2.6)	NA	0.574
**Baseline Psych**					
FSI Disruption	8.1 (11.0)	9.5 (12.4)	8.9 (10.8)	5.9 (9.4)	0.008
STAI State Sum	25.8 (7.3)	26.8 (8.3)	25.5 (6.9)	25.1 (6.6)	0.088
CESD Sum	6.8 (7.1)	7.9 (8.8)	6.6 (6.1)	6.0 (5.9)	0.049

**Table 2 cancers-15-03208-t002:** Results of varying-intercepts and varying-slopes models for the attention, processing speed, and executive function (APE) and learning and memory (LM) domains.

		APE		LM	
Fixed Effects		Mean	95% HDI	Mean	95% HDI
Intercept	$\gamma_{00}$	0.013	−0.110, 0.138	0.048	−0.092, 0.187
**age**	$\gamma_{01}$	**−0.047**	**−0.065, −0.029 ***	**−0.048**	**−0.069, −0.028 ***
age^2^	$\gamma_{02}$	0.00001	−0.002, 0.002	−0.0006	−0.003, 0.002
**survivor**	$\gamma_{03}$	−0.130	−0.279, 0.020	**−0.205**	**−0.376, −0.034 ***
age × survivor	$\gamma_{04}$	0.016	−0.006, 0.038	0.012	−0.013, 0.037
age^2^ × survivor	$\gamma_{05}$	−0.01	−0.004, 0.002	0.0004	−0.003, 0.003
**months ^†^**	$\gamma_{10}$	**0.004**	**0.0004, 0.007 ***	**0.019**	**0.013, 0.025 ***
months × survivor	$\gamma_{11}$	0.0002	−0.004, 0.004	−0.0029	−0.011, 0.005
months × age 69–72	$\gamma_{12.2}$	0.001	−0.004, 0.005	0.0006	−0.008, 0.010
months × age 73–76	$\gamma_{12.3}$	−0.002	−0.008, 0.003	−0.007	−0.016, 0.003
**months × age 77–89**	$\gamma_{12.4}$	**−0.008**	**−0.013, −0.002 ***	−0.007	−0.017, 0.002
survivor × months × age 69–72	$\gamma_{13.2}$	−0.001	−0.006, 0.007	0.0005	−0.011, 0.012
survivor × months × age 73–76	$\gamma_{13.3}$	0.0004	−0.006, 0.007	0.0073	−0.005, 0.019
survivor × months × age 77–89	$\gamma_{13.4}$	0.005	-0.002, 0.011	0.0053	−0.007, 0.017
Random effect. Ω	$\left(\begin{array}{cc} \sigma_{\alpha }=0.60 & \rho =0.16\\ \rho =0.16 & \sigma_{\beta }=0.004 \end{array}\right)$			$\left(\begin{array}{cc} \sigma_{\alpha }=0.64 & \rho =0.27\\ \rho =0.27 & \sigma_{\beta }=0.005 \end{array}\right)$	
Residual error	$\sigma_{\epsilon }^{}$ = 0.194			$\sigma_{\epsilon }$ = 0.361	

^†^ Months refers to months since baseline, as an indicator of practice effect. * Indicates Bayesian 95% credible interval excluding the null (in bold).

## Data Availability

Data will be made available in response to all reasonable requests.

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
