# Peer review of "Cognitive Aging in Older Breast Cancer Survivors"

_cancers, 2023, doi:10.3390/cancers15123208_

Round 1

Reviewer 1 Report

Cognitive aging in older breast cancer survivors
The manuscript, “Cognitive aging in older breast cancer survivors” is an interesting article that should be
published because it helps clarify longitudinal changes in memory and attention over time among breast
cancer survivors and non-cancer controls, and it identifies possible confounds that heretofore clouded
our knowledge in the area of cognitive change over time among cancer survivors. Specifically, the authors
take advantage of interesting methodology using their large data set of 5-15 post-treatment breast cancer
survivors and Bayesian statistical techniques get around the confounds caused by practice effects of
repeat neurocognitive testing, selective attrition of participants in longitudinal studies, and the
impracticality of studies that could take decades to yield meaningful results. The authors used 4 age
cohorts of breast cancer survivors in their large data set that each had 4 testing sessions within about a
2-year timespan. The age cohorts acted as a proxy for life-time trajectory of the data and Bayesian
methods allowed for comparisons of models of trajectories using the cross-sectional testing comparisons
between cancer survivors and controls nested within the cohorts. The main findings were that for the
domains of attention, processing speed and executive function (APE) and Learning and Memory (LM),
younger survivors (in their 60’s) demonstrate greater impairment in these domains but such differences
diminish with time as non-cancer controls decline. The same result occurred for a measure of deficit
accumulation and frailty. The results help elucidate a picture of how cognitive change may occur in
vulnerable cancer patients and this has implications for targeted interventions—both on the preventive
(e.g., lifestyle factors affecting deficit accumulation) and rehabilitation.
Comments below are largely editorial suggestions:
1. Simplify the language in the simple summary. As written, it may not capture the gist of lay or media
readers to summarize the important key findings with the methods reported. For example, change,
“using baseline cognitive performance data as a proxy indicator of potential alterations in cognitive
aging in survivorship,” to perhaps, “we compared different age categories of older breast cancer
survivors with non-cancer participants in the same age categories on standard memory tests to estimate
age and cancer-related memory change over time…” or something to that effect.
2. First sentence in the results section of the abstract is a little confusing given the complexity of the
study design—it states at baseline there were no group differences in APE domain tests, but differences
emerge among the age cohorts later which explains main findings. Perhaps just state the main results in
this part of the abstract.
3. Line 72 p. 2—a confusing couple of sentences. shorten to, "In parallel, recent research found that
longer-term survivors had higher deficit accumulation and worse cognition compared with controls. We
found that cognitive differences between…. Delete “in survivors” at the end of this sentence (line 76)
4. Not sure how adding a non-cancer control group at matched ages accounts for age-cohort effects
(lines 108 to 111 p. 13—may want to revise or clarify).
5. Line 188, shouldn’t this read “pre-frail (DAFI > 0.2, < 0.35)” ?

6. It would be helpful to the reader less familiar with Bayesian statistical methods to explain the
notation or expression of findings that will be seen later in the text. For example, the HDI findings
starting on line 236, p. 6—this can either be briefly stated in the Statistical Approach paragraph or in the
helpful supplemental materials. This would help the reader better understand the magnitudes of effect
and what is statistically meaningful or “significant.
7. Line 236 define “LM”—be sure to be consistent through the rest of the manuscript with this
abbreviation.
8. With the above said, the figures 1, 2 & 3 are extremely helpful.
9. Line 328—typo “we note that that…”
10. Line 394, second sentence. This is an important limitation to report because there are likely minute
or possible cohort effects, such as possible smoking rates when the participants were at younger ages
where there could be cohort variation. Smoking could be a deficit accumulation factor— there is
previous evidence from the investigators in this same sample that early life smoking history may mask
cognitive vulnerability from APOE status. It is recognized that this may be an academic point, but
authors are good to cite this limitation of the current data. Were there any smoking history effects on
cognition within the data presented?

Cognitive aging in older breast cancer survivors
The manuscript, “Cognitive aging in older breast cancer survivors” is an interesting article that should be
published because it helps clarify longitudinal changes in memory and attention over time among breast
cancer survivors and non-cancer controls, and it identifies possible confounds that heretofore clouded
our knowledge in the area of cognitive change over time among cancer survivors. Specifically, the authors
take advantage of interesting methodology using their large data set of 5-15 post-treatment breast cancer
survivors and Bayesian statistical techniques get around the confounds caused by practice effects of
repeat neurocognitive testing, selective attrition of participants in longitudinal studies, and the
impracticality of studies that could take decades to yield meaningful results. The authors used 4 age
cohorts of breast cancer survivors in their large data set that each had 4 testing sessions within about a
2-year timespan. The age cohorts acted as a proxy for life-time trajectory of the data and Bayesian
methods allowed for comparisons of models of trajectories using the cross-sectional testing comparisons
between cancer survivors and controls nested within the cohorts. The main findings were that for the
domains of attention, processing speed and executive function (APE) and Learning and Memory (LM),
younger survivors (in their 60’s) demonstrate greater impairment in these domains but such differences
diminish with time as non-cancer controls decline. The same result occurred for a measure of deficit
accumulation and frailty. The results help elucidate a picture of how cognitive change may occur in
vulnerable cancer patients and this has implications for targeted interventions—both on the preventive
(e.g., lifestyle factors affecting deficit accumulation) and rehabilitation.
Comments below are largely editorial suggestions:
1. Simplify the language in the simple summary. As written, it may not capture the gist of lay or media
readers to summarize the important key findings with the methods reported. For example, change,
“using baseline cognitive performance data as a proxy indicator of potential alterations in cognitive
aging in survivorship,” to perhaps, “we compared different age categories of older breast cancer
survivors with non-cancer participants in the same age categories on standard memory tests to estimate
age and cancer-related memory change over time…” or something to that effect.
2. First sentence in the results section of the abstract is a little confusing given the complexity of the
study design—it states at baseline there were no group differences in APE domain tests, but differences
emerge among the age cohorts later which explains main findings. Perhaps just state the main results in
this part of the abstract.
3. Line 72 p. 2—a confusing couple of sentences. shorten to, "In parallel, recent research found that
longer-term survivors had higher deficit accumulation and worse cognition compared with controls. We
found that cognitive differences between…. Delete “in survivors” at the end of this sentence (line 76)
4. Not sure how adding a non-cancer control group at matched ages accounts for age-cohort effects
(lines 108 to 111 p. 13—may want to revise or clarify).
5. Line 188, shouldn’t this read “pre-frail (DAFI > 0.2, < 0.35)” ?

6. It would be helpful to the reader less familiar with Bayesian statistical methods to explain the
notation or expression of findings that will be seen later in the text. For example, the HDI findings
starting on line 236, p. 6—this can either be briefly stated in the Statistical Approach paragraph or in the
helpful supplemental materials. This would help the reader better understand the magnitudes of effect
and what is statistically meaningful or “significant.
7. Line 236 define “LM”—be sure to be consistent through the rest of the manuscript with this
abbreviation.
8. With the above said, the figures 1, 2 & 3 are extremely helpful.
9. Line 328—typo “we note that that…”
10. Line 394, second sentence. This is an important limitation to report because there are likely minute
or possible cohort effects, such as possible smoking rates when the participants were at younger ages
where there could be cohort variation. Smoking could be a deficit accumulation factor— there is
previous evidence from the investigators in this same sample that early life smoking history may mask
cognitive vulnerability from APOE status. It is recognized that this may be an academic point, but
authors are good to cite this limitation of the current data. Were there any smoking history effects on
cognition within the data presented?

Author Response

Comments below are largely editorial suggestions:

  1. Simplify the language in the simple summary. As written, it may not capture the gist of lay or media readers to summarize the important key findings with the methods reported. For example, change, “using baseline cognitive performance data as a proxy indicator of potential alterations in cognitive aging in survivorship,” to perhaps, “we compared different age categories of older breast cancer survivors with non-cancer participants in the same age categories on standard memory tests to estimate age and cancer-related memory change over time…” or something to that effect.

We have edited the text as suggested: “We compared younger and older breast cancer survivors with younger and older non-cancer controls on standard cognitive measures to estimate age and cancer-related differences over time.”

  1. First sentence in the results section of the abstract is a little confusing given the complexity of the study design—it states at baseline there were no group differences in APE domain tests, but differences emerge among the age cohorts later which explains main findings. Perhaps just state the main results in this part of the abstract.

We have edited the text as suggested:  “Cross-sectional baseline analysis found the expected inverse association of age with cognition in both groups, with survivors overall performing lower than controls in learning and memory (LM). Younger survivors, i.e., those under 75 years of age, exhibited lower performance in both LM and attention, processing speed and executive function (APE) compared to controls, with no differences between older survivors and controls, which tracked with deficit accumulation trends.”

  1. Line 72 p. 2—a confusing couple of sentences. shorten to, "In parallel, recent research found that longer-term survivors had higher deficit accumulation and worse cognition compared with controls. We found that cognitive differences between…. Delete “in survivors” at the end of this sentence (line 76)

We have edited the text as suggested: “In an earlier study, we found that comorbidity burden was higher in patients and associated with baseline cognitive function prior to adjuvant treatment [23]. In parallel, recent research found that longer-term survivors had higher deficit accumulation and worse cognition compared with controls [24], and that cognitive differences were largely mediated by increased deficit accumulation in survivors [25].”

  1. Not sure how adding a non-cancer control group at matched ages accounts for age-cohort effects (lines 108 to 111 p. 13—may want to revise or clarify).

We have edited the text to clarify: “Research from aging literature that disambiguates cohort and aging effects suggests that cohort effects have decreased [27,34] and with the inclusion of a non-cancer control group at matched ages any cohort effects in survivors will be controlled for by contrasts with control participants who are from the same age cohort.”

  1. Line 188, shouldn’t this read “pre-frail (DAFI > 0.2, < 0.35)”?

Yes, thank you for noting this.

  1. It would be helpful to the reader less familiar with Bayesian statistical methods to explain the notation or expression of findings that will be seen later in the text. For example, the HDI findings starting on line 236, p. 6—this can either be briefly stated in the Statistical Approach paragraph or in the helpful supplemental materials. This would help the reader better understand the magnitudes of effect and what is statistically meaningful or “significant.

We have extensively revised the Methods section and the Supplement to address this concern. The Methods section is expanded to explain the notation better, what the key model parameters represent in anticipation for the findings. The Supplement has been revised to contain a brief tutorial to make the technical details more accessible, to explain the “varying intercepts and slopes” approach visually (Supplement Figure 1) with respect to model building. We also cite textbooks that cover Bayesian techniques at a more elementary level and in greater detail, to help researchers begin considering Bayesian statistics in their own work, and to draw readers’ attention to the mini-tutorial.

  1. Line 236 define “LM”—be sure to be consistent through the rest of the manuscript with this abbreviation.

Both APE and LM have been spelled out at first mention in the results section: “Attention, Processing Speed and Executive function (APE) and Learning and Memory (LM) models. . . “

  1. With the above said, the figures 1, 2 & 3 are extremely helpful.

  1. Line 328—typo “we note that that…”

Fixed

  1. Line 394, second sentence. This is an important limitation to report because there are likely minute or possible cohort effects, such as possible smoking rates when the participants were at younger ages where there could be cohort variation. Smoking could be a deficit accumulation factor— there is previous evidence from the investigators in this same sample that early life smoking history may mask cognitive vulnerability from APOE status. It is recognized that this may be an academic point, but authors are good to cite this limitation of the current data. Were there any smoking history effects on cognition within the data presented?

This point is well taken. We have added the following: “As an example, tobacco use has steadily declined in successive generations and previous nicotine exposure has been associated with a potentially protective effect on cognition in APOE4+ carriers [35].”

Reviewer 2 Report

Root et al. presents a clinically relevant study evaluating cognitive performance of breast cancer survivors.  Following issues could be modified:

1. Authors use very complex statistical methods , not standard methodology .  These methods are for many readers new . However the description of methods is not really appropriate. They describe statistical methods using complex statistical language and terms.  Many readers will have big problems to understand methodology of this study.

2. Authors mix differerent study design terms; first they write cross-sectional, then they write cases and controls (would be case-control design in this situation).  They should check which design did they really used and modify terms.  Moreover, did they follow patients from defined index date during defined follow-up?  This would be cohort study then.

3. Consluion: "Differences between survivors and controls were prominent in younger survivors, as was deficit accumulation, suggesting a mediating effect on cognition".  I think, this is important on this place to write which differences they mean?  The most important study result is the difference between breast cancers and non-cancer individuals or even no difference

4. Authors write that they matched patients;  why are sample sizes then so different?

Author Response

  1. Authors use very complex statistical methods , not standard methodology .  These methods are for many readers new . However the description of methods is not really appropriate. They describe statistical methods using complex statistical language and terms.  Many readers will have big problems to understand methodology of this study.

We have expanded the Supplement considerably to serve as a tutorial to help readers understand the equations. As explained in the Supplement, it provides a tutorial on the Bayesian ‘varying slopes and intercepts’ model. It is intended to help readers who are unfamiliar with it to form a more intuitive understanding of the model. It is not intended to be an in-depth summary on Bayesian modeling in general. We also provide further readings to help readers who are unfamiliar with the Bayesian approach, including introductory textbooks that cover the Bayesian approach to hierarchical data, cited in the Supplement. The book by Kruschke is also cited as an accessible tutorial on the basics of Bayesian statistics.

Finally, a brief outline [lines 260 – 267] is added to the Results section, prior to the results in Table 2, to orient readers to the model parameters to aid the interpretation.

  1. Authors mix different study design terms; first they write cross-sectional, then they write cases and controls (would be case-control design in this situation).  They should check which design did they really used and modify terms.  Moreover, did they follow patients from defined index date during defined follow-up?  This would be cohort study then.

We have added the following to clarify the design: “The data reported here was collected as part of a collaboration between Memorial Sloan Kettering Cancer Center and City of Hope (PIs: Ahles; Hurria) on a cohort study that assessed cognition in controls and breast cancer survivors (age 60 or greater) who were all at least 5 years post-treatment and followed prospectively over 2 years with 4 timepoints at 8-month intervals [35].”

One paragraph is added in the Methods section to clarify that the overall study design is a longitudinal observational design with two cohorts, cancer survivors and age- and education-matched non-cancer controls (lines 148 – 149). To address the confusion about the cross-sectional data and longitudinal outcome, we have added one paragraph in the Statistical Approach section (lines 219 – 230) to explain why we use these terms.

  1. Conclusion: "Differences between survivors and controls were prominent in younger survivors, as was deficit accumulation, suggesting a mediating effect on cognition".  I think, this is important on this place to write which differences they mean?  The most important study result is the difference between breast cancers and non-cancer individuals or even no difference

We have clarified that the differences are specifically in LM and APE: “Cognitive differences between survivors and controls for LM and APE domains were prominent in younger survivors, as was deficit accumulation, suggesting a mediating effect on cognition.”

  1. Authors write that they matched patients;  why are sample sizes then so different?

We have added the following to clarify: “Survivor and control groups were frequency matched so that mean age and education were equivalent, but not on a one-to-one basis.”

Round 2

Reviewer 2 Report

N/A